# Ventilator-Associated Pneumonia Prevention in Pediatric Patients: Narrative Review

**DOI:** 10.3390/children9101540

**Published:** 2022-10-09

**Authors:** Natália Antalová, Jozef Klučka, Markéta Říhová, Silvie Poláčková, Andrea Pokorná, Petr Štourač

**Affiliations:** 1Department of Pediatric Anaesthesiology and Intensive Care Medicine, University Hospital Brno and Faculty of Medicine, Masaryk University, Kamenice 5, 625 00 Brno, Czech Republic; 2Department of Health Sciences, Faculty of Medicine, Masaryk University, Kamenice 5, 625 00 Brno, Czech Republic; 3Department of Public Health, Faculty of Medicine, Masaryk University, Kamenice 5, 625 00 Brno, Czech Republic; 4Department of Simulation Medicine, Faculty of Medicine, Masaryk University, Kamenice 5, 625 00 Brno, Czech Republic

**Keywords:** ventilator-associated pneumonia, children, pediatric, intensive care, prevention

## Abstract

Ventilator-associated pneumonia (VAP), one of the most common healthcare-associated infections in intensive care settings, is associated with significant morbidity and mortality. VAP is diagnosed in >10% of patients on mechanical ventilation, incidence rising with number of ventilator days. In recent decades, the pathophysiology of VAP, VAP risk factors and treatment have been extensively studied. In critically ill pediatric patients, mechanical issues such as insufficient tightness of the ventilator circuit (mainly due to historically based preference of uncuffed tubes) and excessive humidity in the circuit are both significant risk factors of VAP development. Protocol-based approaches to critically ill patients on mechanical ventilation, closed suctioning, upper body position, enteral feeding and selective gastric acid suppression medication have a beneficial effect on VAP incidence. In recent decades, cuffed tubes applied to the whole spectrum of critically ill pediatric patients (except neonates <2700 g of weight), together with cuff-oriented nursing care including proper cuff-pressure (<20 cm H_2_O) management and the use of specialized tracheal tubes with subglottic suction ports combined with close infraglottic tracheal suctioning, have been implemented. The aim of this review was to summarize the current evidence-based knowledge about the pathophysiology, risk factors, diagnosis, treatment and prevention of VAP in clinically oriented settings.

## 1. Introduction

Ventilator-associated pneumonia (VAP) is one of the commonest healthcare-associated infections in pediatric ICUs [1]. VAP is defined as a new infiltrate chest X-ray together with positive microbiology findings from the lower respiratory tract and/or two or more positive clinical signs of pneumonia (signs of inflammation, gas exchange impairment) [2,3]. Positive microbiology findings depend on the sampling methods used, where positivity is defined by ≥10^5^ colony-forming units (CFU) in tracheal aspirate, ≥10^4^ in bronchoalveolar lavage and ≥10^3^ in protective specimen brush culture [4]. Despite better understanding of the disease pathophysiology and improvement in prevention methods and therapy, VAP is still a major cause of morbidity and mortality in ICU patients [2,3,5]. According to a 2020 article from Modi and Kovacs [3], 10% of patients put on mechanical ventilation develop VAP and for 13% of those it is a fatal complication. Incidence of VAP in PICUs (pediatric intensive care units) varies significantly around the world, with 1 to 63 episodes per 1000 ventilator days [1,6,7,8]. Moreover, VAP is associated with increased cost and length of hospital stay, increased use of broad-spectrum antibiotics and considerable healthcare costs [9]. Therefore, creating strategies to prevent the development of VAP should be emphasized [10]. Several interventions were previously identified to decrease the incidence of VAP in adult patients; nevertheless, there has been little research into the effects of those bundles on ventilated pediatric patients [5]. Most of the risk factors for developing VAP are known: the most important are enteral feeding and use of certain medication, e.g., proton pump inhibitors or histamine-2 receptor blockers, the first often leading to regurgitation, the latter allowing nosocomial pathogens to colonize the aero-digestive tract and endotracheal tube by suppressing acid production [5,11,12]. Attention must be paid to maintaining tracheal cuff pressure, limiting nonessential tracheal suction or re-intubation, and avoiding overdistention of the stomach or contamination of equipment used [2]. Additionally, patients with genetic syndromes or those suffering from neurological or cardiovascular disease leading to hospital admissions or extended duration of mechanical ventilation carry bigger risks of aspiration [5,11]. Most of the bacteria cultivated from endotracheal aspirates from patients suffering from VAP are also cultivated from naso-oropharyngeal and gastric secretions [2,13]. Based on knowing all the above, the most common methods of prevention in pediatric ICU settings include hand hygiene, mouth rinse with antiseptic solution and elevating the bed to an angle of 30 degrees. Others are changing the ventilator circuit only when it is needed, minimizing ventilation days and draining ventilator condensate away from the patient frequently [2,14].

The promptness of diagnosis is key in prognosis. However, this can be tricky, despite many guidelines, because of the wide spectrum of differential diagnoses for inpatient respiratory declines [3]. The diagnosis is based on the clinical presentation of symptoms that occur after 48 h of mechanical ventilation and require a change in ventilator settings [2]. Many studies agree on new lung infiltrate in chest imaging being the major evidence supporting a VAP diagnosis. Several tests need to be made quickly in order to isolate the etiological pathogen and therefore tailor target antibiotic treatment, meaning sampling sputum and blood for cultures and obtaining PCR-based tracheal sputum (from suctioning or from BAL procedure) and/or nasopharyngeal swabs, together with imaging methods. The imaging method remains the gold standard for VAP diagnosis (together with the pathogen identification and/or signs of inflammation and/or impaired gas exchange), but a possible shift in recent decades could be traced to quick bed-side pneumonia diagnosis via lung ultrasound examination. Despite the use of ultrasound, chest X-rays and lung computed tomography are considered the gold standard. 

As for treatment, generally the first step is initiating broad-spectrum empiric antibiotic therapy. Once the causative pathogen is confirmed, specific antibiotic treatment is chosen to which the pathogen is sensitive (e.g., antibiotic de-escalation) [2]. Common etiological agents are *Pseudomonas* spp., *Staphylococcus aureus* and many Gram-negative bacteria; the proportion and dominance of these pathogenic organisms varies between PICUs over the world [5,15,16].

## 2. VAP Incidence

There have been growing numbers of publications in latest years addressing the need for instituting a set of key evidence-based interventions to fight this preventable hospital-acquired infection (HAI) [2]. The importance of such preventive strategies is underlined by the lack of gold standards for VAP diagnosis. Although many recent studies have demonstrated that implementing VAP prevention bundles could lead to reduced VAP incidence and the importance of such as a daily part of ICU patient care, the appropriate elements that should be included in these prevention strategies have to be evaluated further, mainly for cost–benefit purposes and to assess their efficacy [1]. 

Generally, the incidence of VAP varies, but remains unacceptably high. Differences such as patients’ profiles and nutritional status as well as resource availability and diagnostic criteria can be the source of variation. Pediatric studies across the globe report VAP incidence of 2–35% (of mechanically ventilated patients in PICU) [11,17,18,19,20]. Srinivasan et al. from Boston reported an incidence of 32%, the diagnosis of VAP being established by the CDC [21]. Elward et al. from Missouri then reported a VAP incidence rate of 11.6 per 1000 ventilator days [20]. Vijay et al. documented an incidence rate of 41 episodes per 1000 ventilator days [5]. Amanati al. from Iran than established the incidence rate of VAP in PICUs to 22.9% [22]. According to the reported VAP incidence, it remains the most prevalent healthcare-associated infection in different intensive care settings (neonatal, pediatric, adult) with significant costs, morbidity and mortality worldwide. 

## 3. Diagnosis 

Despite VAP’s significance for clinical practice, there is still no gold standard for a VAP diagnostic. This could have a negative impact on the promptness of VAP management. According to Anita Rae Modi et al.’s article and new guidelines, the diagnosis of VAP requires all the following: new lung infiltrates on chest imaging, respiratory decline, fever and productive cough [3]. If there is no new lung infiltrate on the imaging, the probability of a VAP diagnosis lowers and it is necessary to proceed to thinking about other alternative causes of inpatient respiratory decline [23]. However, it should be noted that in the early stages of pneumonia development, only a few signs of inflammation may be found on the X-ray, and it should be repeated over time in patients with high clinical suspicion (e.g., character of sputum production, cough, fever). Additionally, standard diagnostic criteria should include at least two or three of the following: fever higher than 38 °C or hypothermia lower than 36.5 °C; change in volume or character of sputum or increased need of suctioning; new or worsening cough; breathing problems such as dyspnea, tachypnea or apnea; pathological pulmonary auscultation with signs of rales; and bronchial breath sounds, wheezing or rhonchi. Other criteria are worsening of gas exchange after a period of either improvement on the ventilator or stability, bradycardia or tachycardia (altered hemodynamic), and positive serum biomarkers such as C-reactive protein, procalcitonin or leukocytosis [2].

The diagnostic problem remains that all these clinical criteria have very limited value on their own (not specific for pneumonia itself and have huge overlap with sepsis from other possible etiologies). Many of the signs and symptoms mentioned above may be the consequence of other concurrent morbidities or are routinely present in patients on prolonged mechanical ventilation [2]. On the other hand, VAP clinical presentation could be significantly modified by the concurrent medication often administered to critically ill patients. 

Elsewhere, the CDC—Center for Disease Control and Prevention criteria have been acceptable and used for diagnostic purposes for both pediatric and adult patients. Other criteria used may be Great Ormond Street Hospital (GOSH) criteria or Clinical Pulmonary Infection Score (CPIS). What all these different strategies for diagnosing VAP have in common is the radiographic evidence of pneumonia; what varies are clinical criteria and their respective combinations. Nevertheless, there are difficulties when applying these criteria for VAP, mainly the requirement of radiographic evidence of lung infiltrate and the reliance on subjective clinical symptoms [1].

Taking blood cultures is recommended, as they can be helpful in either revealing the responsible pathogen if respiratory cultures are not conclusive or informing us of the presence of additional infection unrelated to the respiratory tract and therefore are important in differential diagnoses [3].

Acquiring sputum cultures is limited by inpatient ability to provide a sufficient sample of sputum. For patients who are not capable of producing adequate samples, methods such as endotracheal aspiration (this being preferred to the following), bronchoscopy or bronchoalveolar lavage could be chosen instead of simple sputum production by a patient´s active cough. Invasive and quantitative methods may cause patient harm and discomfort and carry no benefits; only in immunocompromised patients or patients non-responsive to therapy could these be appropriate [23,24]. 

PCR testing plays an important part in detecting the responsible pathogen. The nasal swab for *Staphylococcus aureus* demonstrates high negative predictive value for methicillin-resistant *S. aures* colonization and can be used as an antibiotic stewardship tool, prompting safe discontinuation of anti-MRSA therapy when negative [3,25]. Another notable PCR-based test, the nasopharyngeal swab, is used for testing the respiratory viral panel and should be used especially during influenza season, as it identifies the viral origin of the infection and helps in deciding the right therapy [23]. Nowadays PCR testing represents a quick reliable method with high sensitivity to bacteria from the sputum (or tracheal aspirate/BAL). The results of PCR testing could be usable in several hours; however, they could be falsely positive due to high incidence of non-invasive bacterial brands colonizing the airway. The PCR results (always interpreted as quantitative findings—e.g., defining the amount of viral or bacterial load in the selected specimen) should be therefore interpreted together with a patient´s clinical condition. 

Procalcitonin testing helps differentiate viral from bacterial infections, warns us against coinfection and helps decide the duration of therapy. According to Anita Rae Modi et al., the clinical judgement alone is sufficient to initiate antibiotic therapy; however, procalcitonin-guided cessation of antibiotics can be associated with decreased mortality rate [3,26] and reduce the cost of treatment. 

Common etiological agents causing VAP are *Pseudomonas* spp., *Staphylococcus aureus* and various Gram-negative bacilli. The proportion of these organisms varies between the ICUs and PICUs [5]. In addition, we can differentiate between early and late VAP onset that differs in origin, for early-onset *Staphylococcus aureus*, *Streptococcus pneumoniae* and Haemophilus influenzae are typical causative agents. For late-onset VAP, *Pseudomonas* spp., *Acinetobacter* spp. and enteric Gram-negative bacilli are specific. Early-onset pneumonia refers to pneumonia that occurs within four days after mechanical ventilation was initiated, late-onset refers to pneumonia that occurs after more than five days [27]. For example, a prospective cohort study from North India identified these pathogenic organisms as causative agents of VAP, in order of the most frequently isolated: *Acinetobacter*, *Pseudomonas*, *Klebsiella*, *Enterobacter* and *E. coli* [5]. 

## 4. Management

Prompt empiric antibiotic treatment is considered necessary for VAP treatment due to decreasing the chance of sepsis and therefore unfavorable prognosis. Recent studies have argued for the idea that every patient requires immediate antibiotic treatment [3,28]. However, patients who were treated with antibiotics rapidly after showing symptoms could experience longer duration of therapy and higher rate of mortality, probably due to selection of resistant pathogens [28]. Patients who could benefit from immediate and aggressive antibiotic treatment, meaning before the sample cultivation results are available, are those with respiratory or hemodynamic instability, those who are immunocompromised and those for whom it is not possible to appropriately sample respiratory tract secretions [3,23,28]. 

If the decision to treat a patient with suspected VAP has been made, empiric ATB therapy should be initiated. The choice of specific antibiotics should be made with consideration of local hospital bacteria prevalence and resistance. If no such institution specific antibiogram is available, we should resort to empiric coverage of methicillin-susceptible *S. aureus* and Gram-negative bacteria, e.g., piperacillin–tazobactam, cefepime, levofloxacin, imipenem and meropenem [3]. Very often *Pseudomonas aeruginosa* is cultivated from patients who acquired VAP and could be resistant to many of the antibiotic groups. In patients or settings with high possible antibiotic resistance and signs of sepsis, combination therapy should be initiated to cover all potential causative agents. Antibiotics in combination are required in patients who recently received intravenous antibiotic treatment or are at higher risks of mortality as well as those in ICUs where Gram-negative pathogens are often resistant. Ideally, antibiotics from two different classes should be chosen. Patients who are treated with antipseudomonal monotherapy may suffer from delays to the initiation of proper effective antipseudomonal agents [3].

Another question is the empiric coverage of MRSA. Not all patients require therapy with vancomycin or linezolid. However, these antibiotics should be given to patients who received previous intravenous antibiotic therapy in the last 90 days, those hospitalized in a unit where the prevalence of MRSA is either not known or is at least 20% and those who are at high risk of mortality [3,23]. 

Regarding the option of antibiotic class and method of administration, some pathogens causing VAP could be susceptible to aminoglycosides or polymyxins only. These antibiotic classes are known to be nephrotoxic, mainly in their intravenous application. If such situation occurs, according to Anita Rae Modi et al., inhaled aminoglycosides or colistin should be combined with their intravenous formulations [3]. In addition, inhaled forms of these mentioned antibiotics help to achieve higher concentrations at the site of infection that lead to improvement of clinical cure rates as well as reduction of the duration of mechanical ventilation [3,29]. 

Whether it is decided to initiate the empiric antibiotic regimen or not, the course of therapy is then followed by tailoring antibiotic treatment based on the pathogen culture cultivation and its resistances. As is known, aspiration is a major risk factor of acquiring VAP, as aspirates often lead to polymicrobial colonization. Therefore, to treat a patient with suspected aspiration, the final antibiotic regimen should include coverage of oral and enteric flora, inclusive of Gram-negative and anaerobic bacteria [3].

As for the duration of antibiotic treatment, in uncomplicated VAP patients it is seven days. If there are complications such as empyema, bacteremia, pneumonias caused by *Pseudomonas* or *Acinetobacter* species or other pulmonary or extrapulmonary complications, the duration of therapy is longer and specific to the actual complication [3]. 

Generally, it is recommended to use the potentiated betalactams or cephalosporins (e.g., cefotaxime) for early onset VAP and piperacilin–tazobactam for late-onset VAP with addition of aminoglycosides if *Pseudomonas* is suspected [2]. 

## 5. Prevention

Prevention is based mainly on modifying the known VAP risk factors. A list of identified VAP risk factors is mentioned in Table 1. 

Due to the lack of data among pediatric population, the described risk factors are mostly from scientific articles, which interpret this problematic from the adult population. The prevention methods could be divided into two simplified groups: first, the preventive complex care that concerns the caring for of the patient themself by medical staff. That would mean regular hygiene, especially maintaining oral hygiene and hand hygiene, to which the health care staff should strictly comply. Furthermore in this group are the positioning of the patient’s head, a properly selected form of feeding and its administration and medication. All of these mainly concern the prevention of aspiration. The second group would be proper care and management of the equipment—tubes, suctioning techniques, change of ventilation circuits and generally correct processes around the artificial ventilation.

Medical staff’s **hand hygiene** is an important element of preventing health-care-associated infections and spread of multi-resistant pathogens. Incorporation of hand hygiene into the VAP bundle is therefore strongly recommended and different studies have since demonstrated that an adherence to hand hygiene is of relevance to reduction of VAP [2,30,31]. **Oral hygiene** provided to the ventilated patients should be a part of hospital care, as poor oral hygiene has been associated with developing VAP. Colonization of the oral cavity leads to colonization of the tracheal/tracheostomy tube and the lungs by bacteria, which could create easy passage to the lung tissue [2,32,33,34]. There are two main pillars regarding oral hygiene in ventilated patients: Use of mouthwash is one of them, meaning mainly decontamination of the mouth with chlorhexidine solutions. This chemical antiseptic has been suggested as it has the ability to suppress overgrowth of bacteria and yeast as well as reduce dental plaque formation and therefore prevent gingivitis. Hence, after minimizing the colonization of the oral cavity, when micro-aspiration of oral secretions occurs, lung exposure to pathogenic bacteria could be reduced by the effects of chlorhexidine mouthwashes. Secondly, foam swabs or oral toothbrushes are used in order to remove dental plaque and debris. Based on the recommendation, however, dipping these in tap water carries the risk of contamination of pathogenic bacteria and should not be used in critically ill patients to reduce the risk of healthcare-associated infection [2,35,36,37].

Hygiene on the part of the medical staff, as an important component of inpatient care, should be strictly reinforced in the ICU setting as it is essential in preventing VAP.

**Positioning** is considered a significant element in VAP prevention. It is recommended to elevate the head of the bed with the patient to the semi-recumbent position [2,38]. The elevation of the head of the bed should be between 15–30° for neonates and 30–45° for infants or older children according to the article from Marjorie de Neef et al. [9]. Higher risk of aspiration of gastric content to the airway in patients kept in the supine position despite proper inflation of the endotracheal tube cuff was demonstrated in a clinical randomized cross-over trial [2,39].

Additionally, the amount of time that a patient spent in the supine position was proportionate to the increase in aspiration contents [2,40]. Nevertheless, some authors argue with this idea. Panigada et al. claimed that in the semi-recumbent positioning, due to gravitational forces, the contaminated secretions would travel across the tracheal cuff into the lower respiratory tract, potentiated by suctioning, when there is a drop in pressure with the airway. Simultaneously, the lower respiratory tract cannot be cleared of secretions without active suctioning. This was supported by a study on animals that revealed that gravitation influences tracheal mucus clearance after tracheal intubation [2,41,42]. Another study that compared patients in the lateral horizontal position opposite those in the semi-recumbent found comparable incidence of aspiration of gastric secretions in both groups [2,43]. At the moment, there is insufficient high-quality evidence of direct proven advantage of the semi-recumbent position and further studies are needed to evaluate the direct clinical impact of positioning on VAP incidence.

**Enteral feeding** is considered to be another important risk factor for VAP development, as it could lead to regurgitation. We can only assume that implementing prevention methods taking into account all of the above mentioned would help us to improve the care of ventilated patients themselves and decrease the chance of acquiring VAP. Peptic ulcer prophylactics (H2 blockers and proton pump inhibitors—PPI) raise the gastric pH and can lead to increased gastric colonization with pathogenic bacteria and therefore a higher risk of acquiring VAP. The study of Albert et al. [12] revealed increased incidence of VAP with the use of acid-suppression medication. 

As for the use of probiotics in VAP prevention, according to Elias Iosifidiset et al., prophylactic use of probiotics reduced the incidence of VAP by 77% in PICUs [1]. In the study from Anita Rae Modi et al., this effect was explained by the theoretical possibility of intestinal barrier function improvement caused by probiotics, together with the effect of regulating the composition of intestinal flora to minimize over-growth and colonization by pathogenic bacteria and increase host cell antimicrobial peptides [3]. 

Mucolytic agents, such as ambroxol, could be beneficial to ventilated patients regarding the risk of acquiring VAP. This agent has additional antioxidant and anti-inflammatory properties. Besides these features, ambroxol also affects the structure of biofilm formed by *P. aeruginosa* and facilitates the permeability of antibiotics through the biofilm. A study by Fang Li et al. demonstrated that the bacterial counts on the biofilm-covered tubes in patients treated with ambroxol were significantly decreased compared to the control group.. In addition, lower bacterial counts in the lungs as well as milder pathological changes in lungs were found when ambroxol was added to the therapy plan [44]. The results of the study of Yildizdas et al. [45] did not indicate any difference in the incidence of VAP, macroscopic stress ulcer bleeding and mortality between the mechanically ventilated patients hospitalized in the PICU and treated with ranitidine, omeprazole or sucralfate and nontreated subjects. According to the insufficient data about stress ulcer prophylaxis and VAP in the group of pediatric patients, other studies with larger numbers of patients are needed.

Attention should be paid to adequate rehabilitation. Z. Liu et al. studied the effect of rehabilitation on the occurrence of VAP. Although this study worked only with adult patients, it demonstrated that extensive rehabilitation intervention is advantageous to the ICU patients on mechanical ventilation. The incidence of VAP, time on mechanical ventilation and length of hospital stay were lower in patients that received comprehensive rehabilitation treatment (passive exercise therapy, anti-respiratory exercise therapy, active exercise therapy, etc.) compared to patients that were given only routine rehabilitation treatment [27]. 

The implementation of the cuffed tube into the clinical practice seems reasonable as it has the potential to decrease the risk of aspiration and the need for a greater tube exchange rate. In a study comparing the cuffed and uncuffed tubes, it was found that the tracheal exchange rate was lower for the usage of cuffed tubes, with only 2.1% compared to 30.8% in those uncuffed ones [2,46]. As micro-aspirations of oral secretions is one of the biggest risk factors for VAP, using cuffed tubes could be beneficial in prevention considering the superior tracheal seal and its role in lowering the number of micro-aspirations. According to Kneyber et al. the endotracheal high-volume low-pressure cuffed tubes with cuff pressure monitoring could be safely used in all children [47].

There is no increased risk of post-extubation stridor when the cuff pressure monitoring is regularly performed by using cuff-specific devices and the cuff pressure is maintained ≤ 20 cm H_2_O [46,47,48,49]. Despite proper cuff management, the retention of secretion in the subglottic area (over the cuff) could still lead to pericuff leak. This problem could be partially solved with specialized cuffed tubes with the possibility of secretion clearance (by direct suctioning) from the subglottic area—Figure 1.

The use of aseptic techniques while performing endotracheal suction is of great importance to prevent contamination of the respiratory tract [2,50]. This was also highlighted by a study by Sole et. al. (2002), where correlation was found between the bacteria found on respiratory equipment with those in the patient’s mouth and sputum. Based on previously published data, colonization of the suction tubes in the first 24 h by pathogenic bacteria was found to be similar to bacteria in the patient’s mouth and sputum [2,51].

The comparison of open vs. closed suction systems with regard to VAP incidence was conducted by several studies [2]. A closed suction system is preferred because of some specific complications that result from using open system suctioning procedures, such as environmental contamination, cross infections or hypoxia. Despite this obvious disadvantage of open suction systems, no difference in VAP incidence was found utilizing either system [2,52,53]. 

With re-intubation, the risk of VAP becomes greater. An explanation could be that aspiration of gastric contents occurs during the interval between extubation and re-intubation [5]. A different study provides us with a different explanation: entering of bacteria to the lungs directly through the endotracheal tube during disconnection from the ventilator circuit [2]. 

The frequency of ventilator circuit changes should possibly be considered as part of VAP prevention bundles. Several RCTs were conducted to discuss and investigate how frequently humidified circuitry should be changed. All of these studies supported the hypothesis that when the frequency of changing the ventilator circuit was reduced, there was no increased incidence of VAP resulting from that reduction [2,54,55,56,57]. 

Different studies [1,9] mention assessing the readiness to extubate to be part of studied prevention bundles and do not recommend daily interruption of sedation in pediatric modification. The advice against sedation holidays is with consideration of the risk of accidental extubation in young children [9].

Vet et al. said there was no safety issue in daily sedation interruption [58].

All the strategies lead to reducing aspiration risk. One ICU developed a task force and educational session focusing on raising awareness about the importance of aspiration and its prevention, followed by regular assessments of compliance with prevention strategies [59]. The intervention regarding the creation of the task force and educational plan increased compliance significantly and at the same time led to a decrease in VAP by 51%, a lesser amount of ventilation days and decreased healthcare costs. The conclusion here is that use of aspiration-prevention strategies and didactic modules can reduce risk of aspiration and therefore associated pneumonia.

A common issue in bundle preventions is the existence of programs for adult patients but not for pediatric ones. The evidence that prevention bundles can prove effective was given by the 2016 study of De Cristofano et al. They focused on applying a VAP prevention bundle with hope of improving the quality of care in the PICU and decreasing the VAP rate by 25% every 6 months over 2 years. This study settled on four main components of the prevention bundle: head of the bed raise, oral hygiene with chlorhexidine, clean dry ventilator circuit and daily interruption of IV sedatives. Once the bundle was implemented in daily practice, the mean VAP rate dropped from 6.34 episodes every 1000 ventilator days before the intervention to 2.38 [10]. 

The optimal composition of the bundle is not yet known [60]. 

## 6. Evidence-Based Approach and Future Perspective

This review is derived from a number of different articles focused mainly on the assessment of prevention of VAP, its varying incidence, risk factors, diagnosis and need forf prompt treatment. There have been growing numbers of publications in the latest years addressing the need for instituting a set of key evidence-based interventions to fight this preventable healthcare-associated infection [2]. The importance of such preventive strategies is underlined by the lack of gold standards for VAP diagnosis. Although many recent studies have shown that implementing ventilator prevention bundles leads to reduced VAP rates and are important as part of the everyday care of an ICU patient, the appropriate elements that should be included in these prevention strategies have to be evaluated further, mainly for cost–benefit purposes and their efficacy [1].

All the results should be interpreted with respect due to the limitations that all the processed studies carry, that meaning mainly clinical heterogeneity in diagnosis and prevention as well. To conclude, many clinical trials and studies had shown us that implementing a VAP prevention bundle has the potential to reduce VAP incidence in the PICU setting [1]. Therefore, the main focus should be that of implementing those prevention bundles and ensuring compliance with such. However, it should not be forgotten that there is not enough available data to reach definite conclusions regarding the gold standard in both diagnosis and prevention of VAP in children [1]. It would be, therefore, interesting to explore the appropriate elements of such strategies and their effectiveness as well as the educational process that would help us implement preventive bundles and ensure personnel compliance in daily practice.

## 7. Conclusions

Ventilator-associated pneumonia is one of the most prevalent complications of ICU stay and is nowadays still associated with significant morbidity and mortality in critically ill patients. Emphasis should be put mainly on early VAP prevention, possibly by implementing certain “prevention bundles”, together with optimal staff compliance, quick diagnostics and early treatment.

## Figures and Tables

**Figure 1 children-09-01540-f001:**
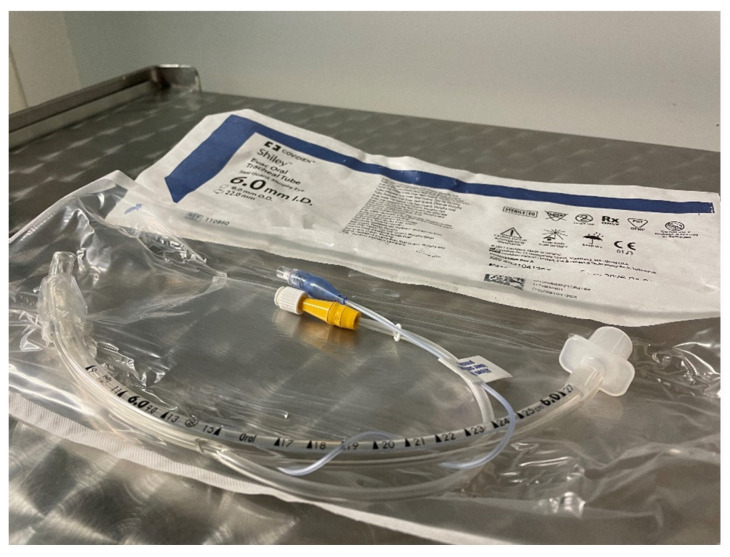
Endotracheal tube with a subglottic suction.

**Table 1 children-09-01540-t001:** Summary of risk factors and preventive measures identified in the literature.

**Risk factors**	Enteral feeding Medication (e.g., proton pump inhibitors, histamine 2-receptor blockers) Re-intubation Aspiration of secretions Use of contaminated equipment Presence of genetic syndrome or neurological/cardiovascular disease Duration of mechanical ventilation
**Preventive measures**	Hand hygiene Mouth rise with antiseptic solution Maintenance of endotracheal cuff pressure of at least 20 cm H_2_O Semi-recumbent position Changing of the ventilator circuit only when visibly soiled Draining ventilator condensate away Minimizing ventilator days

## Data Availability

Not applicable.

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
