# Peer review of "Ventilator-Associated Pneumonia Prevention in Pediatric Patients: Narrative Review"

_children, 2022, doi:10.3390/children9101540_

Round 1
Reviewer 1 Report
This article has no new scientific findings
Author Response
"Please see the attachment."

Reviewer 2 Report
Cuffed tubes are not recommended for neonates, no matter the weight - please modify the paragraph "e. In recent decade, cuffed tubes into the whole spectrum of critically il pediatric patients (except neonates < 2700 grams of weight)......."
CDC developed guidelines for diagnosing VAP in patients younger than 1 year, which include worsening gas exchange, radiographic findings, and at least 3 defined clinical signs of pneumonia - I couldn't find any paragraph about gas exchange in your article ....So please add some info about gas exchange in VAP
Author Response
"Please see the attachment."

Reviewer 3 Report
Dear Authors,
Could you please clarify a few points:
In the introduction, more specifically in the 3rd paragraph, there is a statement that the imaging methods are the Golden standard for the diagnosis of VAP. I think you should review this information because it is inconsistent with the rest of the text and it is also not what is found in the literature where the Golden standard for the diagnosis of VAP is the identification of pathogens in the lower respiratory tract samples through invasive methods such as bronchioalveolar lavage. Because it is invasive, this method is a limiting factor in diagnosing VAP in children.
In the part about VAP incidence, there are quotes about the incidence that it was not clear what the denominator of the rates was. If it was for a patient admitted to the pediatric intensive care unit or for the use of mechanical ventilators.
Furthermore, I would like to suggest that you add other risk factors for VAP in children, since, Knowledge of the risk factors associated with VAP in pediatrics is essential for the introduction of efficient prevention measures. Other risk factors that could be cited are previous use of antibiotics, continuous use of enteral feeding, use of neuromuscular blockers, and prior colonization If possible, you should clarify what are the risk factors found in adult studies and in pediatric studies
Author Response
"Please see the attachment."

Reviewer 4 Report
Thank you for the opportunity to review your manuscript: Ventilator-associated pneumonia prevention in pediatric patients: narrative review
3.DIAGNOSIS
Acquiring sputum cultures is limited by inpatient ability of providing a sufficient 138 sample of sputum. For patients who are not capable of producing adequate samples , 139 methods such as endotracheal aspiration (this being preferred to the later), bronchoscopy 140 or bronchoalveolar lavage could be chosen instead of simple sputum production by pa- 141 tient´s active cough. Invasive and quantitative methods may cause patient’s harm and 142 discomfort and carry no benefits, only in immunocompromised or patients non-respon- 143 sive to therapy, it could be appropriate [22,23]. 144
in this paragraph (3. Diagnosis) I would add some notes on the techniques of sampling from the airways and on the significance and specificity of the same as for example in the example below
Sputum from endotracheal aspirate(>105 CFU/mL) Protected specimen brush (>103 CFU/mL) Bronchoalveolar lavage(>104 CFU/mL)
5.PREVENTION
POSITIONING and Enteral Feeding
something about sucralfate if used in children
Mucolytic agents, such as ambroxol, could be beneficial to ventilated patients re- 290
lower levels of evidence and therefore I would cut the paragraph a lot
Furthermore:
I suggest you reduce the extended form of the text and use summary tables highlighting the level of evidence of the individual phases in both diagnosis, management and prevention.
Author Response
"Please see the attachment."

Round 2
Reviewer 1 Report
It has been added to the knowledge and body of science
Reviewer 4 Report
I thank the authors for considering the suggested corrections. I think the manuscript is clearer now